# Theophylline in Treatment of COVID-19 Induced Sinus Bradycardia

Khalid Sawalha [1,*], Fuad J. Habash [2], Srikanth Vallurupalli [2] and Hakan Paydak [3]

1 Internal Medicine Division, White River Health System, Batesville, AR 72501, USA
2 Cardiology Division, University of Arkansas for Medical Sciences, Little Rock, AR 72501, USA; FHabash@uams.edu (F.J.H.); SVallurupalli@uams.edu (S.V.)
3 Electrophysiology Division, University of Arkansas for Medical Sciences, Little Rock, AR 72501, USA; HPaydak@uams.edu
* Correspondence: Ksawalha@aol.com; Tel.: +1-984-3641-158

**Abstract:** This is a retrospective case series of two patients with laboratory-confirmed coronavirus 2 (SARS-CoV-2) infection, presented to the University of Arkansas for Medical Sciences in January 2021. Medical records of these patients were reviewed using the EPIC electronic health record system. Clinical, laboratory, and treatment data were reviewed against periods of bradycardia in each patient. Both of the patients presented with dizziness and presyncope related to sinus bradycardia in which they received treatment with 1 mg of IV atropine and theophylline 200 mg orally. We share these two cases of theophylline treatment in COVID-19 induced sinus bradycardia. The first patient was a 39-year-old female, with a past medical history of polycystic ovarian syndrome, who presented to the emergency department with lightheadedness and dizziness. Two weeks prior to her presentation, she was tested positive for COVID-19 infection that was treated with azithromycin, dexamethasone and aspirin. Upon presentation, her ECG showed sinus bradycardia at a rate of 48 bpm. The second patient, a 21-year-old female with no significant past medical history, presented with presyncope. Three weeks prior to her presentation, she tested positive for COVID-19 infection that was treated symptomatically at her home. Upon presentation, her ECG showed junctional rhythm at a heart rate of 51 bpm.

**Keywords:** bradycardia; COVID-19; theophylline; bradyarrhythmias; junctional rhythm





## 1. Introduction

Sinus bradycardia is a rhythm in which the rate of impulses arising from the sinoatrial (SA) node is lower than expected. The normal adult heart rate, arising from the SA node, is considered historically to range from 60 to 100 beats per minute, with sinus bradycardia defined as a sinus rhythm with a rate below 60 beats per minute. However, the normal heart rate is, in part, the result of the complex interplay between the sympathetic and parasympathetic nervous systems. It is affected by numerous factors and varies in part with age and physical conditioning [1,2].

Evaluation of beat-to-beat heart rate dynamics, as a result of autonomic nervous system function, is of main interest generally as a higher sympathetic activity unopposed by vagal activity promotes arrhythmia in a variety of ways, such as reducing ventricular refractory period and the ventricular fibrillation threshold, promoting triggered activity afterpotentials and enhancing automaticity. Vagal stimulation opposes these changes and reduces the effects of sympathetic stimulation by prolonging refractoriness, elevating the ventricular fibrillation threshold, and reducing automaticity. Furthermore, the fundamental role of the autonomic nervous system in regulating inflammation, believed to underlie many disease processes, is increasingly being appreciated. Increased sympathetic activity promotes inflammation, and increased vagal activity moderates it [3,4].

The prevalence of arrhythmias and conduction system disease (and cardiovascular disease in general) in patients with COVID-19 varies from population to population. In most available reports, the specific cause of palpitations or types of arrhythmias have not been specified. Hypoxia and electrolyte abnormalities, both known to contribute to the development of acute arrhythmias, have been frequently reported in the acute phase of severe COVID-19 illness; therefore, the exact contribution of COVID-19 infection to the development of arrhythmias in asymptomatic, mildly ill, critically ill, and recovered patients is unknown [3].

## 2. Case 1

A 39-year-old female patient with a past medical history of polycystic ovarian syndrome presented to the emergency department with lightheadedness and dizziness. Two weeks prior to her presentation, she was tested positive for COVID-19 infection after developing symptoms of shortness of breath. She was quarantined at home and treated with azithromycin, dexamethasone and aspirin. She denied smoking and alcohol or illicit drug use. Upon her presentation, her physical examination was noted to be significant for bradycardia. Her vitals were blood pressure of 123/84 mmHg, heart rate of 35 beats per minute (bpm), temperature of 97.7 °F, and a respiratory rate of 20 sating 100% on room air. Her electrocardiogram showed sinus bradycardia without any AV delay or QRS prolongation (Figure 1A). This patient wears a smart watch and she showed us her previous heart rates were in the ranges 70–90 s bpm before her COVID illness.

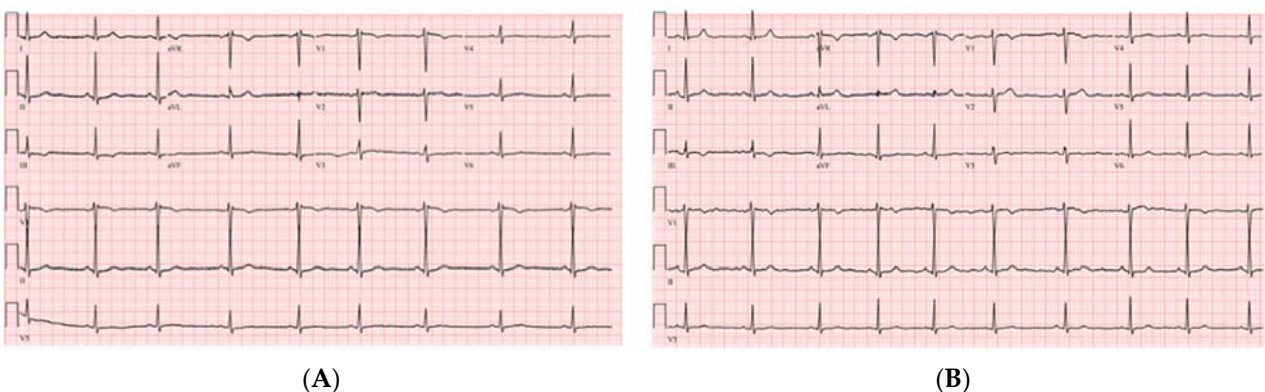

(**A**)          (**B**)

**Figure 1.** The (**A**) electrocardiogram sinus bradycardia at rate of 48 beats per minute, and (**B**) scheme 1 mg of atropine IV with normal sinus rhythm at rate 60 beats per minute.

Significant laboratory results included a WBC of 12.21 k/µL (3.60–9.50), troponin negative at <0.03 ng/mL (≤0.04), normal TSH at 2.65 uIU/mL (0.34–5.60), CRP at 5.9 mg/L (≤10.00), ESR at 36 mm/h (0–20), procalcitonin at <0.02 ng/mL (0.00–0.10), a COVID-19 PCR positive, BNP at 24 pg/mL (≤100), and a negative urine drug screen. The chest X-ray showed no acute events. The transthoracic echocardiogram was normal with an EF of 55%. The patient was given 1 mg of IV atropine with a change of her rhythm to sinus rhythm (Figure 1B). Given her response to atropine, her presentation was likely related to her SARS-CoV-2 infection. She was started on theophylline at a 200 mg oral dose daily. Upon her follow up visit one week later, she reported the improvement of her symptoms with no side effects from the theophylline. Later, at her 6-week follow up, the patient reported complete resolution of her symptoms and, having resumed her normal daily activities and able to exercise normally with her heart rate ranging, per her smart watch, between 100–130 bpm, the decision to discontinue the theophylline was made.

## 3. Case 2

A 21-year-old female patient with no significant past medical history presented to the emergency department with presyncope. Three weeks prior to her presentation, she

tested positive for COVID-19 infection after developing symptoms of shortness of breath and chest pain. She was quarantined at her home and treated symptomatically with improvement. She never required hospitalization for her symptoms nor medications. She denied smoking and alcohol or illicit drug use along with any other medication that could cause QT interval prolongation. Upon her presentation, her physical examination was noted to be significant for bradycardia. Her vitals were blood pressure of 118/63 mmHg, a heart rate of 45 bpm, a temperature of 98.3 °F, and a respiratory rate of 18 sating 100% on room air. Her electrocardiogram showed 45 bpm with narrow QRS complex and premature atrial complexes.

Significant laboratory results included a WBC of 18.24 k/μL (3.60–9.50), troponin negative at <0.03 ng/mL (≤0.04), normal TSH at 3.79 uIU/mL (0.34–5.60), CRP at 27.00 mg/L (≤10.00), ESR at 51 mm/h (0–20), procalcitonin at <0.02 ng/mL (0.00–0.10), a COVID-19 PCR positive, BNP at 160 pg/mL (≤100), lactate at 1.2 mmol/L (0.5–2.2), and a negative urine drug screen. The chest X-ray showed no acute events. The transthoracic echocardiogram was normal with an EF of 60%. The patient was walked on treadmill and her junctional heart rhythm increased to 91 bpm; she was given 1 mg of atropine IV over a period of 5 min, in which her sinus rate increased to >100 bpm. Given the improvement in the patient's symptoms, there was no indication for a dual chamber pacemaker. Hence, she was started on theophylline 200 mg oral dose daily with follow up within 1 week at the clinic (Figure 2A,B). Upon her follow up, she had a heart rate of 90 bpm while on theophylline and no side effects were reported. Later, at her 5-week follow up visit, a complete resolution of her symptoms was reported and theophylline use was discontinued.

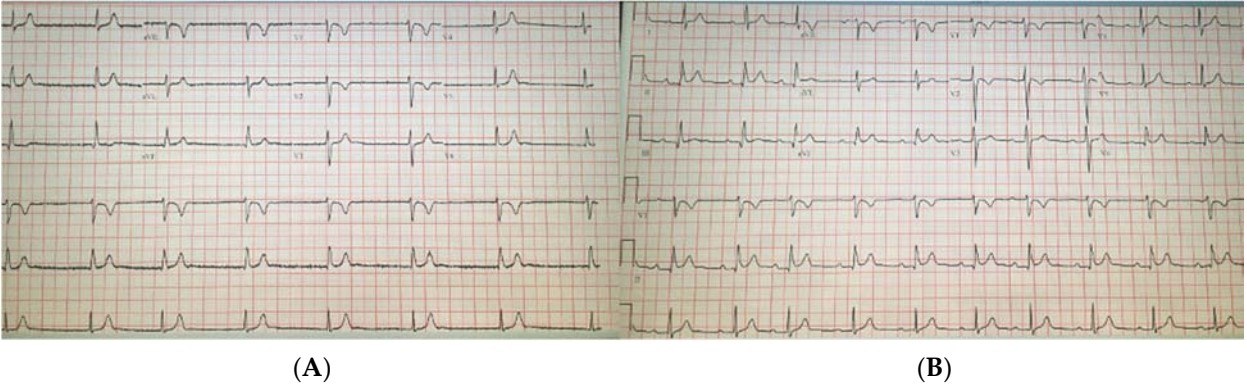

**(A)** **(B)**

**Figure 2.** The (**A**) electrocardiogram at a rate of 45 beats per minute, and (**B**) electrocardiogram with junctional rhythm and resumption of normal sinus rhythm after theophylline.

## 4. Discussion

Normal sinus rhythm (NSR) is the characteristic rhythm of the healthy heart. NSR is considered to be present in adults if the heart rate is between 60 and 100 beats per minute, the P-wave vector on the electrocardiogram is normal (consistent with SA nodal impulse origin), and the rate is mostly regular.

Arrhythmias are most commonly diagnosed from a combination of vital signs and a review of the EKG, ideally a 12-lead EKG, but a rhythm strip can also be used [4].

Sinus bradycardia associated with infection might include viral or bacterial etiologies, such as: Lyme disease, Chagas disease, Legionella, psittacosis, Q fever, typhoid fever, typhus, babesiosis, malaria, leptospirosis, yellow fever, dengue fever, viral hemorrhagic fevers, trichinosis, and Rocky Mountain spotted fever [5,6].

Bradycardia has been reported with COVID-19 infection since the pandemic's start, but to what extent and the exact mechanisms involved are still unknown. One case series on four patients diagnosed with COVID-19 induced bradycardia discussed possible underlying mechanisms, such as: hypoxia, high levels of pro-inflammatory cytokines which may directly affect the sinoatrial node in which bradycardia develops, inflammatory

and catecholamines status that may damage the cardiac pacemaker cells, downregulation of angiotensin-converting enzyme 2 (ACE2) receptor or even medications use [7]. Our patient presented with junctional rhythm in the setting of COVID-19 infection, manifested with an elevation of the ESR and CRP, which indicated that the inflammatory cytokines (mostly through IL-1) released during her immune response, acting on the cardiac pacemaker cells, could possibly have contributed to her presentation.

According to the guidelines of American Heart Association (AHA) and Heart Rhythm Society (HRS), management of patients with asymptomatic bradycardia or pauses does not require placement of a permanent pacemaker [8]. Instead, these patients may be followed with intermittent examinations and observation. However, for patients with symptomatic bradycardia, implantation of a permanent pacemaker, rather than medical therapy or observation alone, is indicated according to the guidelines [8]. Symptoms of syncope and lightheadedness are reversed in all patients following pacemaker placement, but there does not appear to be a survival benefit [9,10].

In a trial of 107 patients with symptomatic sinus node dysfunction who were randomly assigned to no therapy, a rate-responsive pacemaker, or oral theophylline (which can increase heart rate by stimulation of the sympathetic nervous system or blockage of adenosine receptors as adenosine was shown to slow sinus rate and suppress the AV nodal conduction) [11] and followed for an average of 19 months, patients assigned to pacemaker therapy had a significantly lower incidence of syncope compared with those assigned to no therapy (6 versus 23 percent, respectively) and a trend towards less syncope when compared with those receiving theophylline (6 versus 17 percent, respectively) [12]. Implantation of a pacemaker and use of theophylline had an equivalent benefit on the incidence of heart failure compared with controls (3 versus 17 percent, respectively). Therefore, our patients were initiated on theophylline to stimulate their sympathetic system to restore normal sinus rhythm with a dose of 200 mg, taken orally daily, and a titrate as needed based on heart rate response and tolerability. According to the literature, daily doses of up to 900 mg/day have been reported [8]. Patients were educated and counseled about potential side effects of theophylline, including: nausea, vomiting, tachycardia, tremors, headaches, insomnia, and restlessness.

We followed our patients for one week after their discharge to monitor and assess their improvement as well as any potential side effects from the theophylline. No side effects were reported, and a significant improvement was noted. About one month after their initial follow up, another follow up showed the complete resolution of their symptoms and resumption of their daily life activities. Hence, theophylline use was discontinued. This decision was clinically made and driven by the resolution of symptoms as noted in this case series. If there was no significant improvement or persistency of their symptoms, further evaluation would be warranted such as Holter monitoring, stress testing, or even evaluation for pacemaker.

There are limitations to our case series. First, our patients had oxygen saturation 100% on room air which indicates mild pulmonary infection. Other factors may have played a role in bradycardia other than SARS-COV-2, or the infection could have indirectly caused a disruption in the autonomic system balance, such as vagal hyperactivity, situational reflexes, and pain response. Although the patients could have had asymptomatic bradycardia episodes, one of them had a smart watch that showed heart rate trends before hospitalization and she had never had such a low heart rate before the infection. Given that there was no other reversible cause of bradycardia, such as medications or any other obvious reason, we concluded that the symptomatic bradycardia was related to the infection with SARS-COV-2, as it was the only significant change that occurred to both patients. Furthermore, resolution of the infection preceded the resolution of bradycardia-related symptoms and showed a positive heart rate trend during follow up in clinical encounters and vitals taken by the patients themselves at home.

## 5. Conclusions

Our patients, described in this clinical vignette, had signs and symptoms of bradycardia likely induced by their infection with the COVID-19 virus. Treatment with theophylline was proposed previously in the literature for symptomatic patients with sinus node dysfunction, but studies have shown that a pacemaker is superior to theophylline. We used theophylline after the resolution of the patient's symptoms with atropine to restore sinus rhythm as seen in these two cases.

**Author Contributions:** K.S.: writing and gathering information for the manuscript. F.J.H., S.V., and H.P.: reviewing the manuscript. All authors have read and agreed to the published version of the manuscript.

**Funding:** This research received no external funding.

**Institutional Review Board Statement:** Our institution does not require ethical approval for reporting individual cases or case series.

**Informed Consent Statement:** Verbal consent was obtained directly from the patients.

**Data Availability Statement:** Data available upon request.

**Conflicts of Interest:** The authors declare no conflict of interest.

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
