# Peer review of "Theophylline in Treatment of COVID-19 Induced Sinus Bradycardia"

_clinpract, doi:10.3390/clinpract11020047_

Round 1

Reviewer 1 Report

The authors reported two patients whose symptomatic sinus bradycardia probably induced by COVID-19 infection was treated by oral administration of theophylline. The manuscript is well-written and easy to follow. There are several concerns.

  1. How was the long-term outcome following the initiation of theophylline?
  2. Is it possible to further expand the alternative therapeutic tools in addition to theophylline? Or theophylline is the only tool to treat sinus bradycardia induced by COVID-19?

Author Response

Dear reviewer 1,

Thank you for your comments and time, these are excellent comments and I have tried to reply as best I could below. If there are any leading questions, I’ll be happy to take them. Looking forward for your responses. 

The authors reported two patients whose symptomatic sinus bradycardia probably induced by COVID-19 infection was treated by oral administration of theophylline. The manuscript is well-written and easy to follow. There are several concerns.

  1. How was the long-term outcome following the initiation of theophylline?

Reply: Upon following up with the two patients in the clinic one week later, both have reported significant improvement in their symptoms while on Theophylline and resolution of their symptoms on later visits. Theophylline was discontinued after two months.

The authors reported two patients whose symptomatic sinus bradycardia probably induced by COVID-19 infection was treated by oral administration of theophylline. The manuscript is well-written and easy to follow. There are several concerns.

  1. How was the long-term outcome following the initiation of theophylline?

Reply: Upon following up with the two patients in the clinic one week later, both have reported significant improvement in their symptoms while on Theophylline and resolution of their symptoms on later visits. Theophylline was discontinued after two months.

2. Is it possible to further expand the alternative therapeutic tools in addition to theophylline? Or theophylline is the only tool to treat sinus bradycardia induced by COVID-19?

         Reply: Our knowledge of treatment of bradycardia related to COVID is limited and the medical management of bradycardia is also limited. Theophylline is one of the agents that have been used before. Other agents include scopolamine patches and Atropine injections in the acute setting. Definitive treatment includes implantation of pacemakers. In our patient, due to age and unlikely event of damage or injury to the conduction system, we were trying to evade an unnecessary implant and from our case, theophylline was helpful. We do not know the mechanism of bradycardia and its relationship with COVID, but a Chinese paper mentioned presence of ACE reception in the SA node. (Wu J, Deng W, Li S, Yang X. Advances in research on ACE2 as a receptor for 2019-nCoV. Cell Mol Life Sci. 2021 Jan;78(2):531-544. doi: 10.1007/s00018-020-03611-x. Epub 2020 Aug 11. PMID: 32780149; PMCID: PMC7417784.)  

Reviewer 2 Report

Introduction

Lightheadedness 1 word

Case 1

The patient was in sinus rhythm, restoration of SR after atropine does not have sens.

There are missing data about the evolution under Theophylline .

Please use simple past in all sentences.

Case 2

On ECG B there is no junctional rhythm.

Discussion

Per the…      to be replaced  (e.g. according, etc.)

Only the second case is treated.

The most plausible mechanism for COVID induced bradycardia has to be detailed.

The pharmacological properties and mechanism of theophylline has to be detailed.

Conclusions

Please delete the sentence “COVID-19….

Author Contributions – is missing

References

There are some very old items. Please a put a couple of more recent articles.

English gramar needs to be thoroughly revised.

Author Response

Dear reviewer 2,

Thank you for your comments and time, these are excellent comments and I have tried to reply as best I could below. If there are any leading questions, I’ll be happy to take them. Looking forward for your responses. 

Introduction

Lightheadedness 1 word

Reply: Edits are made as requested.

Case 1

The patient was in sinus rhythm, restoration of SR after atropine does not have sense.

There are missing data about the evolution under Theophylline.

Please use simple past in all sentences.

Reply: Edits are made as requested.

Case 2

On ECG B there is no junctional rhythm.

Reply: Edits are made as requested.

Discussion

Per the…      to be replaced (e.g., according, etc.)

Only the second case is treated.

The most plausible mechanism for COVID induced bradycardia has to be detailed.

The pharmacological properties and mechanism of theophylline has to be detailed.

Edits are made as requested.

Conclusions

Please delete the sentence “COVID-19….

Edits are made as requested.

Author Contributions – is missing.

Edits are made as requested.

References 

There are some very old items. Please a put a couple of more recent articles.

English grammar needs to be thoroughly revised.

Edits are made as requested.

Reviewer 3 Report

Attachment

Author Response

Khalid Sawalha et al., describe two clinical cases of COVID-19 affected patients with dizziness and presyncope related to sinus bradycardia, treated with Theophylline 200 mg oral daily.

Bradyarrhythmia’s associated with COVID 19 are common in the clinical practice, and they have different mechanisms such as systemic inflammatory stimulation, hypoxia, inhibitory effect of the virus on the activity of the sinus node, combination or single medications etc. In most cases bradyarrhythmia’s are transient.

Multiple investigational and off-label drugs with are currently used in patients with COVID 19 disease.

Theophylline has relatively good safety profile and could be used to solve symptoms due to temporary bradycardia associated with COVID 19 and avoid PM implantation as suggested by the authors.

Dear reviewer 3,

Thank you for your comments and time. Our knowledge of treatment of bradycardia related to COVID is limited and the medical management of bradycardia is also limited. Theophylline is one of the agents that have been used before. Other agents include scopolamine patches and Atropine injections in the acute setting. Definitive treatment includes implantation of pacemakers. In our patient, due to age and unlikely event of damage or injury to the conduction system, we were trying to evade an unnecessary implant and from our case, theophylline was helpful. We do not know the mechanism of bradycardia and its relationship with COVID, but a Chinese paper mentioned presence of ACE reception in the SA node. (Wu J, Deng W, Li S, Yang X. Advances in research on ACE2 as a receptor for 2019-nCoV. Cell Mol Life Sci. 2021 Jan;78(2):531-544. doi: 10.1007/s00018-020-03611-x. Epub 2020 Aug 11. PMID: 32780149; PMCID: PMC7417784.). Therefore, we are sharing this case series to help with the literature world wide. 

Round 2

Reviewer 2 Report

The issues raised in the review were properly treated. 

Author Response

Thank you sir.